# Photodegradation of Azathioprine in the Presence of Sodium Thiosulfate

**DOI:** 10.3390/ijms23073975

**Published:** 2022-04-02

**Authors:** N’ghaya Toulbe, Ion Smaranda, Catalin Negrila, Cristina Bartha, Corina M. Manta, Mihaela Baibarac

**Affiliations:** 1Laboratory of Optical Processes in Nanostructured Materials, National Institute of Materials Physics, Atomistilor Street 405A, MG-7, 077125 Bucharest, Romania; toulbe.nghaya@infim.ro (N.T.); ion.smaranda@infim.ro (I.S.); 2Interdisciplinary School of Doctoral Studies, University of Bucharest, Șoseaua Panduri 90, 050663 Bucharest, Romania; 3Nanoscale Condensed Matter Laboratory, National Institute of Materials Physics, Atomistilor Street 405A, MG-7, 077125 Bucharest, Romania; catalin.negrila@infim.ro; 4Laboratory of Magnetism & Superconductivity, National Institute of Materials Physics, Atomistilor Street 405A, MG-7, 077125 Bucharest, Romania; cristina.bartha@infim.ro; 5SARA Pharm Solutions S.R.L., 266-268 Calea Rahovei, 050912 Bucharest, Romania; corina.manta@sara-pharm.com

**Keywords:** azathioprine, IR spectroscopy, photodegradation, Raman scattering, UV-VIS spectroscopy, photoluminescence

## Abstract

The effect of sodium thiosulfate (ST) on the photodegradation of azathioprine (AZA) was analyzed by UV-VIS spectroscopy, photoluminescence (PL), FTIR spectroscopy, Raman scattering, X-ray photoelectron (XPS) spectroscopy, thermogravimetry (TG) and mass spectrometry (MS). The PL studies highlighted that as the ST concentration increased from 25 wt.% to 75 wt.% in the AZA:ST mixture, the emission band of AZA gradual downshifted to 553, 542 and 530 nm. The photodegradation process of AZA:ST induced: (i) the emergence of a new band in the 320–400 nm range in the UV-VIS spectra of AZA and (ii) a change in the intensity ratio of the photoluminescence excitation (PLE) bands in the 280–335 and 335–430 nm spectral ranges. These changes suggest the emergence of new compounds during the photo-oxidation reaction of AZA with ST. The invoked photodegradation compounds were confirmed by studies of the Raman scattering, the FTIR spectroscopy and XPS spectroscopy through: (i) the downshift of the IR band of AZA from 1336 cm^−1^ to 1331 cm^−1^, attributed to N-C-N deformation in the purine ring; (ii) the change in the intensity ratio of the Raman lines peaking at 1305 cm^−1^ and 1330 cm^−1^ from 3.45 to 4.57, as the weight of ST in the AZA:ST mixture mass increased; and (iii) the emergence of a new band in the XPS O1s spectrum peaking at 531 eV, which was associated with the C=O bond. Through correlated studies of TG-MS, the main key fragments of ST-reacted AZA are reported.

## 1. Introduction

Azathioprine (AZA) or 6-((1-methyl-4-nitro-1H-imidazol-5-yl)thio)-9H-purine, is an immunosuppressive drug generally used to prevent the rejection of new organ transplants (kidney or liver) [1] or in other medical treatment of some autoimmune diseases (e.g., rheumatoid arthritis [2] and pemphigus [3]), inflammatory bowel illnesses [4] (e.g., Crohn’s diseases, ulcerative colitis), multiple sclerosis and so on [5]. Once in the body, AZA is converted to mercaptopurine, a compound belonging to the class of purines, which will eliminate the effect of some cells in the body’s immune system [6], which may increase the probability of developing an infection [7]. However, many techniques for the determination of AZA both in pharmaceutical formulations and in biological fluids, such as cyclic voltammetry [8], high-performance liquid chromatography [9], flow injection chemiluminescence [10], ^1^H NMR spectroscopy [11], FT-IR spectroscopy [12], Raman scattering [12], surface-enhanced Raman scattering [13] and UV-VIS absorption spectroscopy, ref. [14] have been used. This progress was accompanied by the development of various sensitive platforms for AZA determination such as: (i) hybrid nanocomposites based on graphene oxide functionalized with ionic liquids of type 1-butyl-3-methylimidazolium hexafluorophosphate (GrO-IL) and gold nanoparticles (AuNPs) stabilized by chitosan (Chit) [15]; (ii) Ag nanoparticle-decorated graphene nanosheets [16]; (iii) glassy carbon electrodes modified with a graphene–chitosan (GR-CS) composite [8] and (iv) electrodes based on MgO and multiwalled carbon nanotubes [17].

The diagnosis of skin cancer in patients using AZA after organ transplantation as a post-operative treatment in 1995 [18] stimulated many researchers to become interested in studying the effects of UV exposure on these patients. UV-VIS spectroscopy was used to report on the kinetics of AZA degradation with the assistance of UV light and various oxidant agents such as H_2_O_2_ and sodium persulfate [14,19]. Our previously study concerning the degradation process of AZA induced by O_2_ from air and UV light revealed the emergence of the following two compounds: 4-nitro imidazole 1-carboxylic acid and 6-mercaptopurine [14]. The following spectral variations supported this result: (i) the emergence of the isosbestic point at 320 nm in the UV-VIS spectra; (ii) a change in the PLE bands’ intensity ratio at 275 and 374 nm; (iii) an increase in the intensity of the PL spectra of approximately 111-, 13.5- and 1.45-fold when the excitation wavelength (λ_exc_) was 300, 400 and 450 nm, respectively; (iv) an increase in the IR bands’ absorbance with peaks at 2810 and 3109–3121 cm^−1^ as a result of the emergence of 4-nitro imidazole 1-carboxylic acid; and (v) the emergence of the Raman lines at 1228, 1278, 1350 and 1411 cm^−1^ belonging to 6-mercapto-purine [14]. Y. Zhang et al. demonstrated, by kinetic studies of AZA degradation induced by various oxidative agents, that the most efficient way to remove AZA from natural water was UV light and sodium persulfate [19]. In this research, the effect of the sodium thiosulfate (ST) on the photo-oxidation of AZA was studied. We demonstrate, through correlated studies of the UV-VIS spectroscopy, PLE and PL, that the photo-oxidation of AZA in the presence of ST involves a chemical reaction that leads to the emergence of new C=O functional groups. Some experimental evidence on the photo-oxidation reaction of ST-reacted AZA confirmed the photodegradation pathway proposed in this manuscript by IR spectroscopy, Raman scattering and X-ray photoelectron spectroscopy. The thermal stability of AZA and ST-reacted AZA and the key fragments were highlighted by TG-MS studies.

## 2. Results and Discussion

Figure 1 shows the UV-VIS spectra of the AZA:ST samples in dark conditions and after 6 h of UV irradiation. 

The UV-VIS spectra of AZA:ST in dark conditions, namely before UV irradiation, show a band at 276 nm belonging to the π-π * electronic transition coming from type 1-methyl-4-nitro-1H-imidazol-5yl [20]. As the UV exposure time of AZA:ST increased, we observed a gradual diminution in the band absorbance at 276 nm and the emergence of a band in the 315–405 nm spectral range, which was associated with the n- π * electronic transition belonging to thiopurine and 6-mercapto-purine [21]. Regardless of the ST concentration in the AZA/ST mixture, exposure of the ST-interacted AZA to UV light highlighted the emergence of a band with a maximum around 346–350 nm. Recently, such a band was also reported in the case of the interaction of AZA with H_2_O_2_ [22]. Other changes noted in the UV-VIS spectra of the ST-interacted AZA with an increase in the time of UV exposure were: (i) a downshift of the band from 276 nm (AZA unreacted with ST) to 274 nm (Figure 1a), 270 nm (Figure 1b) and 268 nm (Figure 1c); (ii) an upshift of the band from 346 nm (Figure 1a) to 348 nm (Figure 1b) and 350 nm (Figure 1c); and (iii) the absorbance ratio of the bands in the 245–315 nm and 315–405 nm spectral ranges (A_245-315_/A_315-405_) became equal to 3.2 (Figure 1a), 2.95 (Figure 1b) and 4.22 (Figure 1c) after 6 h of UV irradiation.

Depending on the ST concentration in the AZA:ST mixture, significant variations were observed (Figure 2). According to our previous study, the AZA PLE spectrum is dominated by the two bands in the 280–325 nm and 330–450 nm ranges [14]. As the ST concentration in the ST-reacted AZA samples increased from 25 wt.% to 50 wt.% and to 75 wt.%, the intensity ratio of the two PLE bands above (I_280-325_/I_330-450_) became equal to 1.01, 1.23 and 1.27, respectively. As the UV exposure time of the ST-reacted AZA samples increased up to 6 h, we observed: (i) a gradual increase in the PLE band’s intensity in the 280–325 nm spectral range and an upshift from 282 nm (the AZA sample) to 304 nm (the AZA:ST 25 wt.%: 75 wt.% sample, Figure 2a), 302 nm (the AZA:ST 50 wt.%: 50 wt.% sample, Figure 2b) and 297 nm (the AZA:ST 75 wt.%: 25 wt.% sample, Figure 2c); (ii) a progressive increase in the PLE band’s intensity at 379 nm and 401 nm so that for the AZA:ST samples with an ST concentration in the reaction mixture equal to 25 wt.%, 50 wt.% and 75 wt.%, the PLE band peaked at 401 nm, 400 nm and 392 nm, respectively; and (iii) the intensity ratio of the PLE bands at 280–325 and 330–450 nm (I_280-325_/I_330-450_) of the AZA:ST samples, characterized by a ST concentration in the reaction mixture of 25 wt.%, 50 wt.% and 75 wt.%, became equal to 0.98, 0.82 and 0.77, respectively.

These changes were accompanied by an increase in the intensity of the PL spectra of the AZA:ST samples, as observed in Figure 3. The chemical reaction of AZA with ST induced a downshift of the AZA PL band from 488 nm (2.54 eV) [14] to 492 nm (2.52 eV), 506 nm (2.45 eV) and 503 nm (2.46 eV) for the AZA:ST samples with the ST concentrations in the reaction mixture of 25 wt.%, 50 wt.% and 75 wt.%, respectively (Figure 3). After exposure of the AZA:ST samples with ST concentrations in the reaction mixture equal to 25 wt.%, 50 wt.% and 75 wt.% to UV for time of 10 h, we observed that the maxima of the PL spectra of these samples were situated at 552 nm, 544 nm and 531 nm, respectively, and the values were higher than that of AZA alone (530 nm). Increases in the emission bands’ intensity took place through UV exposure as follows: (i) the AZA:ST 25 wt.%: 75 wt.% sample increased from 1.18 × 10^4^ counts/s to 8.3 × 10^4^ counts/s; (ii) the AZA:ST 50 wt.%: 50 wt.% sample increased from 1.57 × 10^4^ counts/s to 3.43 × 10^5^ counts/s; and (iii) the AZA:ST 75 wt.%: 25 wt.%, sample increased from 1.21 × 10^4^ counts/s to 4.3 × 10^5^ counts/s. These variations indicated increases in the intensity of the PL spectra of the AZA:ST 25 wt.%: 75 wt.%, AZA:ST 50 wt.%: 50 wt.% and AZA:ST 75 wt.%: 25 wt.% samples of ~7-, ~22- and ~35-fold, values which are smaller than that reported for AZA [14].

An explanation for this behavior must take into account the chemical reaction of AZA with ST. To sustain this hypothesis, the IR spectra of ST-reacted AZA (Figure 4) are shown. The IR spectrum of ST shows three bands of high absorbance, peaking at 995 cm^−1^, 1115 cm^−1^ and 1161 cm^−1^, which are in the vicinity of those reported by K. Khairan et al. at 993 cm^−1^, 1109 cm^−1^ and 1156 cm^−1^ [23]. The IR bands at 1652 cm^−1^ and 3400 cm^−1^ are associated with the bending vibration of H_2_O molecules in ST [24,25,26,27]. The IR bands of AZA and the three ST-reacted AZA samples are shown in Table 1.

The main changes observed in Figure 4, as a consequence of the reaction of AZA with ST are: (i) a downshift of the ST IR bands from 1115 cm^−1^ (Figure 4d) to 1113–1107 cm^−1^ (Figure 4b,c) accompanied by a decrease in the absorbance of the IR band in the 3000–3500 cm^−1^ spectral range (Figure 4a–c); (ii) the absorbance ratio of the IR bands at 922–920 cm^−1^ and 1579–1574 cm^−1^ changed from 0.38 (AZA) to 0.94 (AZA:ST 75 wt.%: 25 wt.% sample), 0.95 (AZA:ST 50 wt.%: 50 wt.% sample) and 1.04 (AZA:ST 25 wt.%: 75 wt.% sample); and (iii) the downshift of the IR band of AZA from 1336 cm^−1^ [15] to 1333–1331 cm^−1^ (Figure 4a–c). The downshift of the IR band associated with the N-C-N deformation + C–N stretching + N–H deformation vibration suggests that the reaction of AZA with ST induces changes in the molecular structure of the purine ring. The results which confirm that the purine ring is modified during the reaction of AZA with ST are shown in Figure 5. According to Figure 5d, the Raman spectrum of ST shows an intense line at 436 cm^−1^ with a shoulder at 455 cm^−1^ which is accompanied by other lines of low intensity at 326–347, 553, 675, 995, 1020, 1122 and 1165 cm^−1^. These Raman lines are not far from those reported by M.A. El-Hinnawi et al. at 330–360, 430–450, 530, 670, 1000–1010 and 1125–1140 cm^−1^, which were associated with the vibrational modes of free S_2_O_3_^2−^, symmetric S-S stretching, anti-symmetric S-O bending, symmetric S-O bending, symmetric S-O stretching and anti-symmetric S-O in free S_2_O_3_^2−^ [28]. The Raman lines of AZA are shown in Table 2 [14,29,30].

The reaction of AZA with ST induced the following changes in the Raman spectra of the two compounds: (i) an upshift of the Raman lines of AZA in the 1460–1550 cm^−1^ spectral range; (ii) a change in the intensity ratio of the Raman lines peaking at 1305 cm^−1^ and 1330 cm^−1^ (I_1305_/I_1330_) from 3.45 (AZA) [12] to 4.35 (AZA:ST 75 wt.%: 25 wt.% sample), 4.11 (AZA:ST 50 wt.%: 50 wt.% sample) and 4.57 (AZA:ST 25 wt.%: 75 wt.% sample); and (iii) a change in the intensity ratio of the ST Raman lines at 436 cm^−1^ and 455 cm^−1^ (I_436_/I_455_) from 4.13 in ST to 0.9 (AZA:ST 75 wt.%: 25 wt.% sample), 1.7 (AZA:ST 50 wt.%: 50 wt.% sample) and 1.96 (AZA:ST 25 wt.%: 75 wt.% sample). The variation in the I_1305_/I_1330_ ratio shows that the chemical reaction of AZA with ST induces changes in the purine ring. More conclusive results were reported by XPS spectroscopy.

Figure 6 and Figure 7 show the XPS spectra of ST, and AZA and ST-reacted AZA, respectively. Figure 6 highlights that, by deconvolution: (i) the XPS Na1s spectrum displays two bands at 1071.7 eV and 1073.1 eV (Figure 6a), assigned to ST and sulfate ions, respectively; (ii) the ST XPS Na KLL spectrum displays four bands localized at 990.1, 988.7, 985.9 and 991.6 eV (Figure 6b); (iii) the ST XPS O1s spectrum shows two bands with maxima at 531.7 and 533.5 eV (Figure 6d). The band at 531.7 eV is attributed to the O in sulfites, while the band at 533.5 eV can indicate the O in sulfate and C-O bonds. Returning to Figure 6a,b and considering the bands at 1071.7 eV (Figure 6a) and 990.1 eV (Figure 6b), we can calculate the Auger parameter, which was found to be equal to 2061.9 eV; the value was attributed to sulfites. Figure 6c highlights, in the case of the ST XPS S2p spectrum, two complex bands with maxima at 162.1 and 168.1 eV, and the values are in good agreement with the those reported by B.J. Lindberg et al. [31]. The S2p spectrum was fitted with doublet peaks (S2p^3/2^ and S2p^1/2^) due to spin orbit splitting. The doublet’s separation was 1.2 eV and the peaks were constrained to a 2:1 area ratio during fitting. The bands at 162.1–163.3 eV and 168.1–169.3eV are attributed to Na_2_S_2_O_3_ compounds, as stated in the literature [28,32]. The bands at 166.4–167.6 eV are attributed to the Na_2_SO_3_ compound, while the bands at 169.4–170.6 eV are ascribed to Na_2_SO_4_, as also highlighted in the Na1s spectrum [28,33].

Figure 7a_1_,b_1_,c_1_,d_1_ shows: (i) an AZA XPS C1s spectrum, which, by deconvolution, leads to four bands at 284, 284.8, 285.7, and 286.4 eV, associated with the sp^2^ C=C, C-C + C-H + C-S, C-N/C=N and C-O bonds [34]; (ii) an AZA XPS O1s spectrum, which shows an intense band at 532 eV accompanied by a band of low intensity at 535 eV, associated with the C-O bonds and water traces [34]; (iii) an AZA XPS N1s spectrum that shows three bands at 398.6, 400.3 and 405.3 eV associated with the C=N-C bond [34], the C_2_NH secondary amine [34,35] and the NO_2_ group [36]; and (iv) an AZA XPS S2p spectrum, which highlights, by deconvolution, two bands at 163.4 and 164.6 eV coming from the thiol groups [35].

According to Figure 7a_2_,b_2_,c_2_d_2_, significant changes are induced as a consequence of the reaction of AZA with ST. We noted: (i) the emergence of a new band in (a) the XPS C1s spectrum peaking at 287.7 eV, which was associated with the C=O bond (Figure 7a_2_,b) in the XPS O1s spectrum peaking at 531 eV, which was correlated with the presence of the C=O bond (Figure 7b_2_); (ii) variation in the intensity ratio of the bands at −162.2 eV and 168.2 eV from 0.98 (Figure 6c) to 1.29 (Figure 7d_2_); and (iii) a shift of the band from 1071.7 eV (Figure 6a) to 1072.2 eV (Figure 7e) and the disappearance of the band at 1073.1 eV. An explanation of these changes when AZA reacts with ST—namely (i) the downshift of the IR band of AZA from 1336 cm^−1^ to 1331 cm^−1^, the band attributed to the N-C-N bond’s deformation vibration in the purine ring; (ii) the change in the I_1305_/I_1330_ ratio from 3.45 to 4.57 as the ST weight in the AZA:ST mixture mass increased; and (iii) the emergence of the band in the XPS O1s spectrum peaking at 531 eV, which was associated with the C=O bond—must consider Figure 1.

The reaction products shown in Figure 1 are sustained by: (i) the downshift of the IR band of AZA from 1336 cm^−1^ to 1331 cm^−1^, attributed to N-C-N deformation in the purine ring, as a consequence of the transformation of the -N=CH-NH- bonds into the -NH-C(O)-NH- bonds; and (ii) the emergence of a new band in the XPS O1s spectrum peaking at 531 eV, which was associated with the C=O bond.

The TG-DSC curves corresponding to the AZA sample (Figure 8a) highlight its thermal stability up to 249 °C. The thermal decomposition occurring after this temperature takes place in two steps. In the first step (247–270 °C), a loss in mass of about 2.2% is observed on the TG curve. This is accompanied by a succession of two visible peaks on the DSC curve. The first is a small endothermic peak at 250 °C followed by an exothermic one with a higher intensity, centered at 269 °C. The second step of the thermal decomposition 270–500 °C has a mass loss of ~7.2% and is accompanied by a succession of endothermic peaks of low intensity (DSC curves) due to oxidation effects, considering that the sample was measured in synthetic air. The thermal decomposition behavior of Na_2_S_2_O_3_ is shown in Figure 8b. The first two endothermic peaks (DSC curves) with maxima at 58 °C and 95 °C, respectively, are most likely due to the process of evaporation of sulfur and of some water molecules present on the surface. These are accompanied by two mass losses of 6.1% (the first step) and 4.43% (the second step) on the TG curve. Over the range from 123 °C to 140 °C, there is a large mass loss (~17.1%) accompanied by an endothermic peak centered at 137 °C. This effect can be assigned to thermal degradation of ST [37,38,39]. Due to the fact that the sample was analyzed in synthetic air, in the sequence of reactions specific to the thermal decomposition of Na_2_S_2_O_3_, an oxidizing effect of the released sulfur may also occur. The endothermic peak at 335 °C can be assigned to such a process [37,38,39]. The sequence of the next three exothermic peaks with maxima at 403 °C, 440 °C and 465 °C was associated with the vaporization of the gas mixture resulting from thermal decomposition, especially elemental sulfur. This hypothesis is supported by the boiling point of sulfur being 444.6 °C [37,38,39]. Figure 8c shows the thermal behavior of the AZA after UV irradiation. In this case, we observed an exothermic peak at 259 °C, succeeded by thermal decomposition in the range of 270–500 °C. Figure 8d shows the thermal behavior of the ST-reacted AZA (AZA:ST 50 wt.%: 50 wt.% sample). The thermal behavior of this sample sums up the effects identified in the two samples described above (AZA and Na_2_S_2_O_3_). For the thermal decomposition carried out over the entire measured range (RT-500 °C), the TG curve registered a continuous decrease in a sequence of seven steps. The total loss of mass was ~14%. One conclusion would be that the chemical reaction between AZA and ST generates better thermal stability than that of each separate analyzed compound. For the first time for these compounds, mass spectrometry was used to perform a qualitative analysis of the generated gas components. The data were acquired with MID (Multiple Ion Detection) measurements. In this method, the resulting gases were calculated by multiplying the intensities by previously determined factors [40]. Those calibration factors were determined by performing a gas-specific calibration measurement. In our work, the MS data were acquired through calibration of the ambient air species (argon, nitrogen and oxygen). The recorded MS curves are shown in Figure 9. The possible gases and/or molecular fragments resulting from the pyrolysis of the four samples are presented in Table 3. The key fragments were identified using the NIST database library [41].

The TG-DSC curves showed that the thermal decomposition of the samples is a complex process and takes place throughout the entire measured range (25–500 °C). The MS curves (Figure 9) confirmed this fact by continuously recording the gases and/or molecular fragments resulting from these decompositions. The almost linear aspect of the MS curves is due to the rather low concentrations of the compounds resulting from the thermal degradation of the samples. A more concrete estimation of these is highlighted in Table 3 by the percentages of relative abundance. The large number of the resulting compounds is justified by the complex chemical reaction mechanisms specific to the analyzed materials. Most of the recorded gases are energy-stable and were identified with the abovementioned database. However, in the thiosulphate spectrum, there were two molecular fragments (m/e = 40 and m/e = 47, respectively) that could not be identified. They most likely correspond to metastable compounds obtained from the reaction of the sample with the working atmosphere (synthetic air). The two possible fragments are sodium hydroxide (m/e = 40) and nitrous acid (m/e = 47).

## 3. Materials and Methods

AZA (C_9_H_7_N_7_O_2_S, purity ≥ 98%), ST (Na_2_S_2_O_3_ × 5 H_2_O, purity ≥ 99.5%) and ethanol (C_2_H_6_O, purity ≥ 99.8%) were purchased from Sigma Aldrich (St. Louis, MO, USA). The ST-reacted AZA photodegradation was monitored by UV-VIS spectroscopy, PL, FTIR spectroscopy, Raman scattering and XPS spectroscopy.

To study the impact of ST on the photodegradation of AZA by UV-VIS spectroscopy, the following solutions of AZA and ST in C_2_H_5_OH were prepared: (i) 1 mL AZA (1.52 × 10^−4^ M) and 1 mL ST (5.6 × 10^−5^ M); (ii) 1 mL AZA (10^−4^ M) and 1 mL ST (1.12 × 10^−^^4^ M); and (iii) 1 mL AZA (10^−6^ M) and 1 mL ST (3.38 × 10^−4^ M). The three solutions were further mixed under ultrasonication for a time of 2 min.

For the PL, FTIR spectroscopy, Raman scattering and XPS spectroscopy analyses, three samples were prepared by mixing of AZA and ST in the solid state. The resulting samples of AZA:ST had concentrations equal to: (i) 25 wt.%: 75 wt.%, (ii) 50 wt.%: 50 wt.% and (iii) 75 wt.%: 25 wt.%, t labeled as Samples A, B and C.

The UV-VIS spectra of the AZA:ST samples were registered with a Lambda 950 UV-VIS-NIR spectrophotometer, purchased from Perkin Elmer (PerkinElmer, Inc., Waltham, MA, USA).

The IR spectra of the three samples of AZA:ST were registered with a Vertex 80 FTIR spectrophotometer from Bruker (Billerica, MA, USA). 

The Raman spectra of the AZA:ST samples were recorded at λ_exc_ = 1064 nm with a RFS100S FT Raman spectrophotometer from Bruker (Ettlingen, Germany).

The XPS spectra of ST, AZA and AZA:ST were achieved with a SPECS spectrometer (SPECS Gmbh, Berlin, Germany) endowed with a PHOBIOS 150 analyzer and an Al Kα source.

For the studies using UV-VIS and FTIR spectroscopy, Raman scattering and XPS spectroscopy, the photodegradation of AZA:ST was achieved by UV irradiation at 253 nm under a 350 W mercury-vapor lamp.

The PL and PLE spectra of the AZA:ST samples were measured with a Flurolog-3 spectrophotometer (FL3-221 model) from Horiba Jobin Yvon (Palaiseau, France), endowed with a Xe lamp of 500 W. 

The thermal analysis experiments were undertaken with a SETARAM Setsys Evolution 18 Thermogravimeter (Caluire-et-Cuire, France) (Al_2_O_3_ crucibles) in TG-DSC mode in the range of 20–500 °C. Samples with initial mass of ≈11 mg were measured in synthetic air (20%O_2_; 80%N_2_) with a flow gas rate of 16 mL/min. The heating rate was 5 °C/min. The accuracy of the heat flow measurements was ±0.001 mW and the temperature precision was ±0.01 °C. The analysis by mass spectrometry was performed by monitoring the gases with a Setaram QMS 301 Omnistar Pfeiffer mass spectrometer (Pfeiffer-Vacuum.com, Berlin, Germany), which was coupled to the Setaram Setsys Evolution 18 equipment.

## 4. Conclusions

In this article, new results concerning the reaction of AZA and ST assisted by UV light were reported. By using UV-VIS spectroscopy, photoluminescence, IR spectroscopy, Raman scattering and X-ray photoelectron spectroscopy, it was demonstrated that the photochemical reaction of AZA with ST induces: (i) the emergence of an isosbestic point at 340 nm, as a consequence of the emergence of new photodegradation products; (ii) a decrease in the I_280-320_/I_330-450_ ratio in the PLE spectra from 0.98 to 0.77 as the ST weight in the AZA:ST reaction mixture increased; (iii) the PL band’s intensity increased by ~7-, ~22- and ~35-fold when the AZA:ST samples had an ST concentration equal to 25 wt.%, 50 wt.% and 75 wt.%, respectively (these values are smaller than those reported for AZA alone, as a result of the reaction of AZA with ST); (iv) the IR spectroscopy and Raman scattering analyses highlighted the changes in the purine ring’s vibrational modes, as a consequence of the reaction of AZA with ST; (v) using XPS spectroscopy, the emergence of a band appearing to the C=O bond was reported; this result supported the idea that the proposed chemical reaction occurred when AZA reacts with ST; and (vi) the chemical reaction of AZA with ST generates a better thermal stability than that of each separate analyzed compound.

## Data Availability

Data is contained within the article.

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
