# Peer review of "Photodegradation of Azathioprine in the Presence of Sodium Thiosulfate"

_ijms, 2022, doi:10.3390/ijms23073975_

Round 1

Reviewer 1 Report

The authors presented Photodegradation of azathioprine in the presence of sodium 2

thiosulfate.  These is interesting data and the article in a good production and can be accepted after minor revision but after considering the following points 

My comments

1- All materials used in the investigation should be mentioned with their purity

2- For assignments of IR, the following citations should be added to update ref. 23

Computational Biology and Chemistry 97, 2022,  107643; Journal of Molecular Structure 1242, 2021, 130693; Arabian Journal of Chemistry 10, 2017, S1835-S1846

3- It is more convenient to make a comparison with other related studies in literature

4- The authors must revise language of the manuscript before publication and the whole article should be adjusted based on journal style.

Author Response

My comments

1- All materials used in the investigation should be mentioned with their purity

 Authors reply: In the revised manuscript, this information has been included at line 276 as follows:“AZA (C9H7N7O2S, purity ³ 98%), ST (Na2S2O3 x 5 H2O, purity ³99.5%) and ethanol (C2H6O, purity ³99.8%) were purchased from the Aldrich Sigma company. “

2- For assignments of IR, the following citations should be added to update ref. 23 Computational Biology and Chemistry 97, 2022, 107643; Journal of Molecular Structure 1242, 2021, 130693; Arabian Journal of Chemistry 10, 2017, S1835-S1846

 Authors reply: In the revised manuscript, above articles are find as References 25, 26 and 27.

3- It is more convenient to make a comparison with other related studies in literature

 Authors reply: In the revised manuscript, we have included the following comment at:- lines 95-98:“Regardless of the ST concentration in the AZA/ST mixture, the exposure to UV light of the ST-interacted AZA highlights the emergence of a band with the maximum around 346-350 nm. Recently, such a band was also reported in the case of the interaction of AZA with H2O2 [22]. “- lines 167-173:“According to Figure 5d, the Raman spectrum of ST shows an intense line at 436 cm-1 having a shoulder at 455 cm-1 which is accompanied of other lines of low intensity at 326-347, 553, 675, 995, 1020, 1122 and 1165 cm-1. These Raman lines are not far from those reported by M.A. El-Hinnawi et al., i.e., at 330-360, 430-450, 530, 670, 1000 - 1010 and 1125 - 1140 cm-1, which were associated to the vibrational modes of free S2O32-, symmetric S-S stretching, anti-symmetric S-O bending, symmetric S-O bending, symmetric S-O stretching and anti-symmetric S-O in free S2O32- [28]. “-lines 138-145:“An increase of the emission bands intensity takes place by the UV exposure of: i) the AZA:ST 25 wt.% : 75 wt.% sample, from 1.18 x 104 counts/sec to 8.3 x 104 counts/sec; ii) the AZA:ST 50 wt.% : 50 wt.% sample, from 1.57 x 104 counts/sec to 3.43 x 105 counts/sec; and iii) the AZA:ST 75 wt.% : 25 wt.%, sample, from 1.21 x 104 counts/sec to 4.3 x 105 counts/sec. These variations indicate an increase of the PL spectra intensity of the AZA:ST 25 wt.% : 75 wt.%, AZA:ST 50 wt.% : 50 wt.% and AZA:ST 75 wt.% : 25 wt.% samples of ~7, ~22 and ~35 times, values which are smaller than that reported for AZA [14].  “

4- The authors must revise language of the manuscript before publication and the whole article should be adjusted based on journal style.

Authors reply: In the revised manuscript, a check of English language has been performed. Some examples are:- Figures 6 and 7: “Bending Energy (eV)” has been replaced with “Binding Energy (eV)”;-line 92: “As the UV exposure time increasing of AZA:ST….” has been rewritten as follows “ As increase the UV exposure time of AZA:ST …..“;-line 95: “Regarless of the ST concentration……” has been rewritten as follows “Regardless of the ST concentration …. “;-lines 102-104: “Other changes remarked as the time of UV exposure increases up to 6 hours in Figure 1 are:….. “ has been rewritten as follows “Other changes remarked in spectra UV-VIS of the ST-interacted AZA with increasing the time of UV exposure are:….. “;-line 241: “An explanation for these changes, during the reaction of AZA with ST, …. “ has been rewritten as follows “An explanation for these changes, when AZA reacts with ST, …. “.The section entitled “Results and discussion “ is shown between lines 82-313. 

Reviewer 2 Report

The work "Photodegradation of azathioprine in the presence of sodium thiosulfate" is devoted to the study of photolysis of azathioprine in the presence of sodium thiosulfate. The team of authors is actively studying the mechanisms of azathioprine photodegradation and has repeatedly published their results in prestigious journals. In the current work, the authors focused on the presence of sodium thiosulfate in the reaction mixture and tried to determine the mechanism of the reaction and the resulting products. Photolysis products of the reaction mixture of azathioprine and sodium thiosulfate were analyzed using UV-Vis absorbance and fluorescence spectroscopies, FRIR spectroscopy, XPS and TG-MS.

Unfortunately, the design of the work does not allow for a reliable assessment of the results, since the work is presented as a raw report. After demonstrating the results of the analysis methods, the authors mainly list the observed results, without paying attention to discussion and explanations. From the presented results it is difficult to judge the reliability of the conclusions that the authors make at the end of the work. Also, the abundance of raw experimental results worsens the perception of the work. In my opinion, tables and most of the spectra, from which conclusions significant for the work do not follow, can be transferred to supporting information. I hope that the authors will be able to refine the current manuscript, after which it can probably be accepted into the IJMS journal.

Detailed questions.

  1. Figure 1. What does the red line correspond to?
  2. Lines 94-96. The authors discuss the isosbestic point, but the spectral range around 340 nm changes during the irradiation, which is probably accompanied by a chemical reaction. In my opinion, the use of this term here is incorrect.
  3. Line 107 and further. In my opinion, the notation a, b and C is unclear for samples of reaction mixtures. 
  4. FTIR and Raman spectra from figures 4 and 5 will be more informative if combined on the same graph. It is also desirable to indicate the observed changes in these spectra depending on the concentration of thiosulfate.
  5. On graphs, do not use a comma in numbers.
  6. Sign the peaks on the XPS spectra. Indicate not energies, but atoms types, oxidation state or bonds.
  7. 2 decimal places in binding energies are excessive precision for XPS.
  8. Lines 183-184 Calculated Auger parameter of sodium corresponds to Na+. Other reasoning may not be correct, since based on NIST database is most close to 2061.86 eV are Na2SeO3 or NaCl, that’s wrong. 
  9. Fig 6. The authors do not draw any conclusions and do not give reasoning about the spectra of oxygen and sulfur. What about the spectrum of carbon?
  10. Scheme 1. Prepare a scheme of molecules in suitable software.
  11. Scheme 1. The proposed reaction scheme does not explicitly follow the presented results. Please complete the text with an explanation proving the presented scheme.
  12. Mass spectrometry results (Figure 9 and Table 3) are not discussed in the text. It is not clear integral values of ion currents or results at a fixed temperature are given. The accuracy that is given is redundant. Generally, TG-MS is not a quantitative method for analyzing thermal degradation products without performing calibrations using external standards. In addition, the mass spectrum is not completely deciphered, 
  13. The text also mentions Na2S2O8. If this is a typo, correct it.
  14. The text contains many typos that need to be corrected.

Author Response

Please, see attached document.

Round 2

Reviewer 2 Report

After manuscript improvement, this article satisfies for publication in the IJMS. In my opinion, most of the tables need to be moved to supplementary materials, but the decision remains with the authors.